# Lifetime and past-year suicidal behaviors among adolescents in Bangladesh: A two-stage stratified cluster sampling study

Firoj Al-Mamun[1,2,3] ⓘ, Abdullah Al Habib[4] ⓘ, Johurul Islam[1,5] ⓘ, Moneerah Mohammad ALmerab[6] ⓘ, Mohammed A. Mamun[1,2,3] ⓘ and Mohammad Muhit[1,5] ⓘ

[1]Department of Public Health and Informatics, Jahangirnagar University, Dhaka, Bangladesh; [2]CHINTA Research Bangladesh, Savar, Dhaka, Bangladesh; [3]Department of Public Health, University of South Asia, Dhaka, Bangladesh; [4]Department of Government and Politics, Jahangirnagar University, Savar, Dhaka, Bangladesh; [5]CSF Global, Banani, Dhaka, Bangladesh and [6]Department of Psychology, College of Education and Human Development, Princess Nourah Bint Abdulrahman University, Riyadh, Saudi Arabia

## Research Article

**Keywords:**
suicidal behavior; adolescents; depression; high school students; Bangladesh

**Corresponding author:**
Mohammed A. Mamun;
Email: mamun@thechinta.org

## Abstract

Adolescence is a critical period marked by significant physical and psychological changes, yet there is limited understanding of suicidal behaviors among adolescents in Bangladesh. To address this gap, the MeLiSA study utilizing a two-stage stratified cluster sampling approach was conducted to investigate the prevalence and associated factors of suicidal ideation, plans and attempts among adolescents. A total of 1,496 participants were recruited from urban and rural areas, and their socio-demographic characteristics and data on smoking, alcohol use, depression, anxiety and insomnia were obtained. Chi-square and Fisher's exact tests were used for univariate analyses, followed by multivariable logistic regression to identify factors associated with suicidal behaviors. The findings revealed that 6.8% reported experiencing lifetime suicidal ideation, with 2.3% suicide plans and 0.8% suicide attempts. The 12-month prevalence rates were 3.2% for suicidal ideation, 1.5% for suicide plans and 0.6% for suicide attempts. Smoking emerged as a significant predictor of suicidal ideation, plans and attempts, while alcohol use was strongly linked to past-year suicide attempts. Depression was associated with lifetime suicidal ideation, whereas anxiety significantly influenced both lifetime and past-year suicide plans. These results provide valuable insights that could inform evidence-based interventions and policies to address prevalent mental disorders and suicidal behaviors among adolescents in Bangladesh.

## Impact statement

The findings of this study have significant implications for a range of stakeholders, including policymakers, educators, mental health professionals and the general public. By providing a detailed analysis of suicidal behaviors, including suicidal ideation, plans and attempts, among adolescents in Bangladesh, this study highlights the urgent need for targeted interventions to address the mental health crisis faced by this vulnerable population. Adolescents are particularly vulnerable due to rapid physical, cognitive and emotional changes they experience during this developmental stage, which can increase susceptibility to mental health challenges. *Locally*, the study offers valuable insights for school administrators, healthcare providers and community leaders, who can collaborate with mental health experts to develop and implement effective mental health programs and support systems tailored to the unique needs of adolescents. By identifying key risk factors such as age, rural residency, smoking, depression and anxiety, the study provides a foundation for creating targeted prevention and intervention strategies that can reduce suicidal behaviors. *Regionally*, these findings can guide public health initiatives and inform policies aimed at enhancing mental health support in underserved areas. The evidence presented calls for improved resource allocation and the integration of mental health services into educational and community settings to better address the needs of adolescents. *Internationally*, this study contributes to the global discourse on adolescent mental health by providing critical data on a previously underrepresented population. It highlights the importance of culturally and contextually relevant interventions and offers a model for similar studies in other regions with comparable challenges. *Overall*, this study emphasizes the need for comprehensive and proactive approaches to adolescent mental health, with the potential to drive meaningful changes and improve well-being both locally and globally.

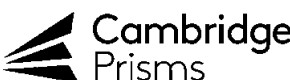

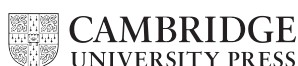

## Introduction

Suicide is the fourth leading cause of death among individuals aged 15–29, claiming over 700,000 lives annually, making it a significant global public health concern (WHO 2021). Suicidal behaviors encompass a range of actions and thoughts related to self-harm or the intention to end one's life, including suicidal ideation (thinking about suicide), suicide attempts and completed suicide (Turecki and Brent 2016; WHO 2021). According to the WHO (2021), 77% of suicides occur in low- and middle-income countries. In Bangladesh, a troubling trend has been observed. In 2022, 446 students died by suicide, with 340 being school-going students, as reported by the Aachol Foundation (Dhaka Tribune 2022). A study conducted during the COVID-19 pandemic revealed that 42.5% of the suicide victims involved individuals aged 14–18 years, with the mean age of them reported as 16.44 (±3.512) years (Mamun et al. 2023). Recent data from January to August 2023 indicate that 361 students died by suicide; among them, 46.8% were school students, 26.6% were college students and 18.3% were university students (Dhaka Tribune 2023). With an average of 45.13 suicides per month over the past 8 months, there is an urgent need for targeted interventions and support mechanisms within Bangladesh's educational institutions (Dhaka Tribune 2023).

Adolescence is a critical developmental stage bridging childhood and adulthood, marked by significant physical, cognitive and psychosocial changes (Sawyer et al. 2018). These transformations can impact emotions, cognitive processes, decision-making abilities and interpersonal relationships, potentially leading to suicidal behaviors (WHO 2023). Numerous studies have explored suicidal behaviors among adolescents. For instance, in the United States, a nationwide school-based survey found that 18.8% of students experienced suicidal ideation, 15.7% had a suicide plan and 8.9% attempted suicide (Ivey-Stephenson et al. 2020). Similarly, research among 728 school-going adolescents in Chile revealed a suicidal ideation rate of 65.6% and a suicide attempt rate of 18.4%, with aggressiveness and bullying identified as significant contributing factors (Veloso-Besio et al. 2023). A recent meta-analysis of studies from 90 countries, involving 327,299 adolescents, identified bullying, lack of close friendships, previous serious injuries and involvement in physical altercations as key factors contributing to suicidal ideation (Campisi et al. 2020).

In Bangladesh, research on adolescents (defined as ages 10–19, as suggested by the WHO [2023]), suicidal behavior remains sparse, with only a few studies exploring this critical issue in depth. For example, a 2013 cross-sectional study involving 2,476 adolescents and young adults aged 14–19 years in rural areas reported that 5% of participants experienced lifetime suicidal ideation and age, education, occupation, living arrangements and house ownership were identified as the significant factors of suicidal ideation (Begum et al. 2017). In addition, two other studies utilized data from the 2014 Global School-Based Student Health Survey. One of these studies, which combined data from Bangladesh and Indonesia, assessed the prevalence and risk factors of suicidal behavior but did not provide specific insights into the Bangladeshi context (Marthoenis and Arafat 2022). The other study reported a prevalence of 11.7% for suicidal behaviors in Bangladesh and identified a range of associated risk factors, including feelings of loneliness, anxiety, bullying, lack of close friendships, engagement in risky behaviors (e.g., sexual activity, alcohol and drug use) and socio-environmental factors like infrequent parental homework check and inadequate peer support (Khan et al. 2020).

While these studies provide valuable insights, they exhibit limitations that our study seeks to address. Specifically, they rely on data from approximately a decade ago (e.g., data from 2013 and 2014), lack differentiation between urban and rural contexts, and focus primarily on general demographic and behavioral risk factors (Begum et al. 2017; Marthoenis and Arafat 2022; Khan et al. 2020). Our study expands upon these analyses by including factors often overlooked in the Bangladeshi context, such as depression, insomnia and specific lifestyle behaviors. Furthermore, we offer an updated, comprehensive approach that examines all aspects of suicidal behavior (e.g., ideation, plan and attempts) together. Furthermore, this study integrates culturally relevant factors such as parental educational background, family structure and rural vs. urban residency, which are especially significant in understanding the social dynamics and vulnerabilities of Bangladeshi adolescents. Hence, this study aims to bridge these gaps by offering an updated, in-depth analysis of the prevalence and associated factors of suicidal behaviors among adolescents in both urban and rural areas. Aligned with the National Adolescent Health Strategy (2017–2030), this study is crucial for developing targeted, evidence-based interventions by shedding light on the magnitude and factors underlying adolescent suicidal behaviors, thereby informing effective nationwide policies.

## Methods

### Study design, participants, procedure and ethics

A cross-sectional study was carried out in November 2022 among high school students in Shahzadpur Upazila, Sirajganj district, one of the 64 districts in Bangladesh. The study employed a two-stage stratified cluster sampling technique to ensure a representative sample of the population. In the first stage, three urban and four rural schools were randomly selected as the primary sampling units, reflecting the diverse geographical settings of the district. In the second stage, students from randomly selected grades (7, 8 and 9) within each school were identified and all students enrolled in these selected grades were invited to participate in the study, ensuring representation across the selected age groups, typically 12–17 years old.

Before data collection commenced, approval was secured from the relevant school authorities and class teachers, ensuring institutional support and compliance with ethical standards. Additionally, written informed consent was obtained from the students, which required parental or guardian review and approval for participation in the study. Data were collected in group settings within classrooms, under the supervision of two investigators (FAM and MAM) and implemented by three research assistants. This team was responsible for administering the classroom-based survey, clarifying any doubts, ensuring the confidentiality of all responses and addressing any concerns raised by both the teachers and the students.

The inclusion criteria for participation in the study were as follows: (i) being present during the survey period and (ii) being enrolled in grades 7, 8 or 9. Students who did not provide consent, had physical disabilities that could interfere with their participation, were absent during the survey, or provided incomplete information on the outcome variable were excluded from the study to maintain the integrity and accuracy of the findings.

The study was conducted in accordance with the ethical principles of the Helsinki Declaration, 2013. Ethical approval for the study was granted by the University of South Asia, Dhaka, Bangladesh after evaluation of the research proposal. Written informed consent was obtained from school authorities, parents and all participants,

ensuring they were fully aware of the study's purpose, procedures and their right to withdraw at any time. To ensure the participant confidentiality, all data were anonymized and securely stored throughout the study. Moreover, we implemented a protocol allowing participants to withdraw from the study if they felt distressed, with immediate support available.

### Measures

#### Sociodemographic and lifestyle factors

The sociodemographic factors assessed in this study include age (categorized as 12–14 years vs. 15–17 years), gender (boys vs. girls), grade (7, 8 or 9), location (urban vs. rural), number of family members (categorized as <5, 6–10 and >10), family type (joint vs., nuclear), birth order (categorized as first, second and third or more), BMI (height and weight), and both parental education (primary or less, secondary and tertiary) and occupation. BMI was calculated as $kg/m^2$ and categorized as underweight, normal, overweight or obese. Father's occupation was categorized as employment, business and others; and service and housewife were the category for mother's occupation. Lifestyle-related questions were posed, focusing on whether participants had ever smoked or consumed alcohol, with responses recorded in a binary format (Yes/No).

#### Mental health problems

Depression was assessed using the Bangla nine-item Patient Health Questionnaire (PHQ-9; Kroenke et al. 2001; Rahman et al. 2022). Participants rated items such as "Little interest or pleasure in doing things" on a four-point Likert scale (0 = not at all, 1 = several days, 2 = more than half of the days and 3 = nearly every day). Scores ranged from 0 to 27, with a cutoff score of ≥10 indicating depression, which has an 88% sensitivity and 88% specificity for screening purposes (Kroenke et al. 2001). While it was initially developed for adults, research has demonstrated its validity and reliability in adolescent populations (Andreas and Brunborg 2017; Johnson et al. 2002; Richardson et al. 2010). The Bangla version of the scale was previously used in Bangladeshi samples (Nahrin et al. 2023; Rahman et al. 2022). A complete version of the scale, including all items and response options, is provided in the Supplementary Materials.

Insomnia was evaluated using the two-item Insomnia Severity Index (ISI-2; Kraepelien et al. 2021). Participants responded to items like "Satisfied/dissatisfied with current sleep pattern" on a five-point Likert scale (0 = very satisfied to 4 = very dissatisfied). Scores ranged from 0 to 8, with a cutoff of ≥6 for insomnia. This cutoff has a sensitivity of 84% and a specificity of 76% (Kraepelien et al. 2021). The Bangla version of the scale was previously used in Bangladeshi samples (Hasan et al. 2021). A complete version of the scale, including all items and response options, is provided in the Supplementary Materials.

Anxiety related to generalized anxiety disorder was measured using the seven-item Bangla Generalized Anxiety Disorder (GAD-7) scale (Dhira et al. 2021; Spitzer et al. 2006). Items such as "Feeling nervous, anxious or on edge" were rated on a four-point Likert scale (0 = not at all, 1 = several days, 2 = more than half of the days and 3 = nearly every day). Scores ranged from 0 to 21, with a cutoff of ≥10 indicating anxiety, which has a sensitivity of 89% and a specificity of 82% for screening (Spitzer et al. 2006). The Bangla version of the scale was previously used in Bangladeshi samples (Dhira et al. 2021; Nahrin et al. 2023). A complete version of the scale, including all items and response options, is provided in the Supplementary Materials.

#### Suicidal behaviors

To assess suicidal behaviors, participants were asked about their thoughts on dying by suicide, whether they had made plans to do so, and if they had attempted suicide. They used a binary response scheme (Yes/No). This method aligns with established concepts of suicidality and ensures consistency and comparability with evaluation methods used in previous studies (Mamun et al. 2022b; Turecki and Brent 2016). The questions addressed both lifetime and past-year occurrences of suicidal behaviors.

### Statistical analysis

The data were first entered, cleaned and made ready for analysis using Microsoft Excel 2021. After that Statistical Package for Social Science version 25 was used for data analysis. Chi-square and Fisher's exact tests were conducted to examine associations between participant characteristics and lifetime and past-year suicidal behaviors (suicidal ideation, suicide plans and suicide attempts). Fisher's exact test was used where the expected count less than five was more than 20%. To identify the associated factors, a multiple logistic regression was applied considering all the variables in terms of lifetime and past year suicide behaviors. The model fit was assessed by the Hosmer–Lemeshow test where $p > 0.05$ indicates a good model fit. There were no missing cases for major variables such as depression, anxiety, insomnia and suicidal behaviors. Missing data were minimal and primarily associated with secondary demographic variables, which accounted for less than 2% of the dataset. These cases were excluded from the analysis, and missing data in the logistic regression was handled using the listwise deletion method. All the test results were significant at $p < 0.05$ with a 95% confidence interval.

### Results

#### Description of the study participants

The study included 1,496 adolescents from Shahzadpur Upazila, Sirajganj district, Bangladesh, with a mean age of 13.91 years (SD = 0.02). The sample was almost evenly split between boys (807, 53.9%) and girls (689, 46.1%). The sample was stratified into two age groups: 12–14 years (72.1%) and 15–17 years (27.9%). Participants were enrolled in grades 7 (27.1%), 8 (42.6%) and 9 (30.2%). The unequal distribution reflects the enrollment sizes of grades in the selected schools. The study encompassed both urban (49.1%) and rural (50.9%) participants. Socio-demographic characteristics such as number of family members, family type, birth order, BMI and parental education and occupation were also recorded (Table 1).

#### Prevalence of suicidal behavior

The results revealed that the prevalence of lifetime suicidal ideation, suicide plans and suicide attempts was 6.8%, 2.3% and 0.8%, respectively. In the past year, the rates were 3.2% for suicidal ideation, 1.5% for suicide plans and 0.6% for suicide attempts.

#### Univariate associations between study variables and suicide ideation

Lifetime suicidal ideation varied significantly across several variables. The prevalence was higher among adolescents aged 15–17 years compared to those aged 12–14 years (9.4% vs. 5.8%; $\chi^2 = 6.18$, $p = 0.013$). Gender differences showed that girls had a higher rate of lifetime suicidal ideation compared to boys (8.9% vs. 5.1%;

**Table 1.** Characteristics of the participants and their associations with suicidal ideation

| Variables | Total, n (%) | Lifetime suicide ideation | | | Past year suicidal ideation | | |
|---|---|---|---|---|---|---|---|
| | | Yes, n (%) | $\chi^2$ test value | p-value | Yes, n (%) | $\chi^2$ test value | p-value |
| Age group (mean ± SD = 13.91 ± 1.02) | | | | | | | |
| 12–14 | 1,075 (72.1) | 62 (5.8) | 6.18 | **0.013** | 32 (3) | 0.389 | 0.533 |
| 15–17 | 416 (27.9) | 39 (9.4) | | | 15 (3.6) | | |
| Gender | | | | | | | |
| Boys | 807 (53.9) | 41 (5.1) | 8.32 | **0.004** | 20 (2.5) | 3.00 | 0.083 |
| Girls | 689 (46.1) | 61 (8.9) | | | 28 (4.1) | | |
| Grade | | | | | | | |
| 7 | 395 (27.1) | 6 (1.5) | 24.20 | **<0.001** | 3 (0.8) | 10.61 | **0.005** |
| 8 | 620 (42.6) | 49 (7.9) | | | 27 (4.4) | | |
| 9 | 440 (30.2) | 42 (9.5) | | | 15 (3.4) | | |
| Location | | | | | | | |
| Urban | 734 (49.1) | 89 (12.1) | 63.88 | **<0.001** | 47 (6.4) | 47.35 | **<0.001** |
| Rural | 762 (50.9) | 13 (1.7) | | | 1 (0.1) | | |
| Number of family members | | | | | | | |
| <5 | 1,070 (72.7) | 76 (7.1) | 0.75 | 0.687 | 38 (3.6) | 3.22 | 0.199 |
| 6–10 | 368 (25) | 22 (6) | | | 7 (1.9) | | |
| >10 | 34 (2.3) | 3 (8.8) | | | 2 (5.9) | | |
| Family type | | | | | | | |
| Nuclear | 1,205 (83.3) | 89 (7.4) | 1.79 | 0.181 | 39 (3.2) | 0.004 | 0.947 |
| Joint | 241 (16.7) | 12 (5) | | | 8 (3.3) | | |
| Birth order | | | | | | | |
| First | 751 (52.9) | 54 (7.2) | 2.54 | 0.281 | 29 (3.9) | 4.54 | 0.103 |
| Second | 402 (28.3) | 24 (6) | | | 8 (2) | | |
| Third or more | 267 (18.8) | 12 (4.5) | | | 5 (1.9) | | |
| BMI | | | | | | | |
| Underweight | 585 (39.9) | 23 (3.9) | 12.35 | **0.002** | 6 (1) | 14.67 | **0.001** |
| Normal | 721 (49.2) | 62 (8.6) | | | 30 (4.2) | | |
| Overweight or obese | 159 (10.9) | 14 (8.8) | | | 9 (5.7) | | |
| Father's education | | | | | | | |
| Primary or less | 452 (34.9) | 14 (3.1) | 32.12 | **<0.001** | 5 (1.1) | 36.53 | **<0.001** |
| Secondary | 542 (41.9) | 37 (6.8) | | | 13 (2.4) | | |
| Tertiary | 301 (23.2) | 42 (14) | | | 27 (9.0) | | |
| Mother's education | | | | | | | |
| Primary or less | 428 (32.5) | 12 (2.8) | 39.12 | **<0.001** | 5 (1.2) | 31.09 | **<0.001** |
| Secondary | 729 (55.4) | 51 (7) | | | 24 (3.3) | | |
| Tertiary | 160 (12.1) | 28 (17.5) | | | 17 (10.6) | | |
| Father's occupation | | | | | | | |
| Employed | 367 (25.2) | 30 (8.2) | 25.12 | **<0.001** | 16 (4.4) | 17.31 | **<0.001** |
| Business | 524 (36) | 54 (10.3) | | | 27 (5.2) | | |
| Others | 566 (38.8) | 16 (2.8) | | | 5 (0.9) | | |
| Mother's occupation | | | | | | | |
| Service | 129 (8.7) | 20 (15.5) | 17.16 | **<0.001** | 10 (7.8) | 9.62 | **0.002** |
| Housewife | 1,351 (91.3) | 80 (5.9) | | | 37 (2.7) | | |

(*Continued*)

**Table 1.** (*Continued*)

| Variables | Total, n (%) | Lifetime suicide ideation | | | Past year suicidal ideation | | |
|---|---|---|---|---|---|---|---|
| | | Yes, n (%) | $\chi^2$ test value | p-value | Yes, n (%) | $\chi^2$ test value | p-value |
| **Ever had smoking** | | | | | | | |
| No | 1,380 (95) | 86 (6.2) | 22.54 | **<0.001** | 39 (2.8) | 20.03 | **<0.001**\* |
| Yes | 72 (5) | 15 (20.8) | | | 9 (12.5) | | |
| **Ever had alcohol** | | | | | | | |
| No | 1,433 (99) | 98 (6.8) | 4.54 | 0.068\* | 47 (3.3) | 0.645 | 0.378\* |
| Yes | 14 (1) | 3 (21.4) | | | 1 (7.1) | | |
| **Depression** | | | | | | | |
| No | 1,240 (82.9) | 44 (3.5) | 121.94 | **<0.001** | 20 (1.6) | 59.40 | **<0.001** |
| Yes | 256 (17.1) | 58 (22.7) | | | 28 (10.9) | | |
| **Insomnia** | | | | | | | |
| No | 1,460 (97.6) | 90 (6.2) | 40.82 | **<0.001**\* | 44 (3) | 7.41 | **0.026**\* |
| Yes | 36 (2.4) | 12 (33.3) | | | 4 (11.1) | | |
| **Anxiety** | | | | | | | |
| No | 1,390 (92.9) | 68 (4.9) | 114.55 | **<0.001** | 32 (2.3) | 51.89 | **<0.001**\* |
| Yes | 106 (7.1) | 34 (32.1) | | | 16 (15.1) | | |

\*Fisher's exact test was applied for categorical variables with low expected cell frequencies (<5).
Bold values indicate statistically significant results (*p* < 0.05).

$\chi^2 = 8.32$, $p = 0.004$). Grade level also impacted ideation rates, with the highest prevalence observed in grade 9 (9.5%), followed by grade 8 (7.9%) and grade 7 (1.5%); ($\chi^2 = 24.20$, $p < 0.001$. Urban adolescents reported a much higher rate of lifetime suicidal ideation compared to their rural counterparts (12.1% vs. 1.7%; $\chi^2 = 63.88$, $p < 0.001$). Lower BMI categories were associated with higher rates of ideation ($\chi^2 = 12.35$, $p = 0.002$), and lower parental education levels were significant factors (father's education: $\chi^2 = 32.12$, $p < 0.001$; mother's education: $\chi^2 = 39.12$, $p < 0.001$). Adolescents with depression had a significantly higher prevalence of ideation compared to those without depression (22.7% vs. 3.5%; $\chi^2 = 121.94$, $p < 0.001$). Similarly, participants suffering from insomnia ($\chi^2 = 40.82$, $p < 0.001$) and anxiety (4.9%; $\chi^2 = 114.55$, $p < 0.001$) were highly reporting suicidal ideation (Table 1).

Past-year suicidal ideation was similarly influenced by several factors. Gender differences were observed, with girls showing a higher rate (4.1%) compared to boys (2.5%), though this difference was not statistically significant ($\chi^2 = 3.00$, $p = 0.083$). Grade level affected the past-year ideation rates, with 4.4% in grade 8 and 3.4% in grade 9, compared to 0.8% in grade 7 ($\chi^2 = 10.61$, $p = 0.005$). Urban adolescents had a significantly higher rate of past-year suicidal ideation compared to rural adolescents (6.4% vs. 0.1%; $\chi^2 = 47.35$, $p < 0.001$). Additionally, those with depression (10.9% vs. 1.6%) and anxiety (15.1% vs. 2.3%) reported higher rates of past-year suicidal ideation compared to those without these conditions ($p < 0.001$ for both). Insomnia also showed a significant association, with those experiencing insomnia reporting higher rates of past-year suicidal ideation compared to those without insomnia (11.1% vs. 3.0%; $\chi^2 = 7.41$, $p = 0.026$; Table 1).

### Univariate associations between study variables and suicide plans

Lifetime suicide plans were significantly associated with several variables. Adolescents in grade 8 had the highest prevalence

(3.4%), compared to grades 9 (1.8%) and 7 (0.5%) ($\chi^2 = 9.89$, $p = 0.007$). Urban adolescents reported substantially higher rates compared to their rural counterparts (4.5% vs. 0.1%; $\chi^2 = 32.06$, $p < 0.001$). Similarly, higher rates of lifetime plans were observed among adolescents with tertiary-educated parents (fathers: 7.0%; mothers: 7.5%) compared to those with parents having lower education levels ($p < 0.001$ for both). Adolescents with depression and anxiety were significantly more likely to report lifetime plans than those without these conditions ($p < 0.001$ for both). Smoking also showed a strong association, with adolescents who smoked reporting higher rates of lifetime plans compared to nonsmokers (9.7% vs. 2.0%; $\chi^2 = 18.04$, $p < 0.001$; Table 2).

Past-year suicide plans followed similar trends. Urban adolescents reported suicide plans, whereas no participants in rural areas did so ($\chi^2 = 23.18$, $p < 0.001$). Grade 8 students again reported the highest rates compared to other grades ($\chi^2 = 11.72$, $p = 0.003$). Depression, anxiety, and smoking were all significantly associated with past-year plans ($p < 0.001$). Parents' higher education levels (fathers: 4.7%; mothers: 6.3%) also contributed to higher prevalence rates ($p < 0.001$; Table 2).

### Univariate associations between study variables and suicide attempts

Lifetime suicide attempts were significantly higher among urban adolescents compared to rural adolescents (1.5% vs. 0.1%; $\chi^2 = 8.78$, $p = 0.003$). Grade 8 students had the highest prevalence compared to other grades ($\chi^2 = 5.40$, $p = 0.067$). Smoking, alcohol use and depression were strongly associated with lifetime attempts ($p < 0.05$ for all). Additionally, adolescents with tertiary-educated fathers (1.7%) and mothers (1.9%) showed higher rates of lifetime attempts than those with less educated parents (Table 3).

**Table 2.** Associations between the study variables and suicide plans

| Variables | Lifetime suicide plan | | | Past year suicide plan | | |
|---|---|---|---|---|---|---|
| | Yes, *n* (%) | $\chi^2$ test value | *p*-value | Yes, *n* (%) | $\chi^2$ test value | *p*-value |
| Age group | | | | | | |
| 12–14 | 26 (2.4) | 0.75 | 0.386 | 18 (1.7) | 1.96 | 0.161 |
| 15–17 | 7 (1.7) | | | 3 (0.7) | | |
| Gender | | | | | | |
| Boys | 15 (1.9) | 1.35 | 0.245 | 9 (1.1) | 1.52 | 0.217 |
| Girls | 19 (2.8) | | | 13 (1.9) | | |
| Grade | | | | | | |
| 7 | 2 (0.5) | 9.89 | **0.007** | 0 (0) | 11.72 | **0.003** |
| 8 | 21 (3.4) | | | 15 (2.4) | | |
| 9 | 8 (1.8) | | | 4 (0.9) | | |
| Location | | | | | | |
| Urban | 33 (4.5) | 32.06 | **<0.001** | 22 (3) | 23.18 | **<0.001** |
| Rural | 1 (0.1) | | | 0 (0) | | |
| Number of family members | | | | | | |
| <5 | 26 (2.4) | 0.87 | 0.645 | 16 (1.5) | 0.89 | 0.640 |
| 6–10 | 6 (1.6) | | | 4 (1.1) | | |
| >10 | 1 (2.9) | | | 1 (2.9) | | |
| Family type | | | | | | |
| Nuclear | 30 (2.5) | 1.39 | 0.237 | 18 (1.5) | 0.08 | 0.768 |
| Joint | 3 (1.2) | | | 3 (1.2) | | |
| Birth order | | | | | | |
| First | 17 (2.3) | 1.32 | 0.515 | 10 (1.3) | 1.19 | 0.550 |
| Second | 8 (2.0) | | | 7 (1.7) | | |
| Third or more | 3 (1.1) | | | 2 (0.7) | | |
| BMI | | | | | | |
| Underweight | 6 (1) | 7.09 | **0.029** | 2 (0.3) | 8.31 | **0.016** |
| Normal | 21 (2.1) | | | 16 (2.2) | | |
| Overweight or obese | 6 (3.8) | | | 3 (1.9) | | |
| Father's Education | | | | | | |
| Primary or less | 2 (0.4) | 32.92 | **<0.001** | 3 (0.7) | 24.91 | **<0.001** |
| Secondary | 10 (1.8) | | | 3 (0.6) | | |
| Tertiary | 21 (7) | | | 14 (4.7) | | |
| Mother's education | | | | | | |
| Primary or less | 3 (0.7) | 22.04 | **<0.001** | 3 (0.7) | 25.42 | **<0.001** |
| Secondary | 18 (2.5) | | | 8 (1.1) | | |
| Tertiary | 12 (7.5) | | | 10 (6.3) | | |
| Father's occupation | | | | | | |
| Employed | 12 (3.3) | 10.77 | **0.005** | 8 (2.2) | 5.98 | **0.050** |
| Business | 18 (3.4) | | | 11 (2.1) | | |
| Others | 4 (0.7) | | | 3 (0.5) | | |
| Mother's occupation | | | | | | |
| Service | 7 (5.4) | 7.11 | **0.008** | 6 (4.7) | 10.55 | **<0.001** |
| Housewife | 25 (1.9) | | | 15 (1.1) | | |

(*Continued*)

**Table 2.** (*Continued*)

| Variables | Lifetime suicide plan | | | Past year suicide plan | | |
|---|---|---|---|---|---|---|
| | Yes, *n* (%) | $\chi^2$ test value | *p*-value | Yes, *n* (%) | $\chi^2$ test value | *p*-value |
| Ever had smoking | | | | | | |
| No | 27 (2) | 18.04 | **<0.001** | 18 (1.3) | 8.28 | **0.004** |
| Yes | 7 (9.7) | | | 4 (5.6) | | |
| Ever had alcohol | | | | | | |
| No | 32 (2.2) | 8.77 | **0.003** | 21 (1.5) | 2.98 | **0.084** |
| Yes | 2 (14.3) | | | 1 (7.1) | | |
| Depression | | | | | | |
| No | 15 (1.2) | 36.86 | **<0.001** | 9 (0.7) | 27.74 | **<0.001** |
| Yes | 19 (7.4) | | | 13 (5.1) | | |
| Insomnia | | | | | | |
| No | 29 (2) | 22.41 | **<0.001** | 19 (1.3) | 11.99 | **<0.001** |
| Yes | 5 (13.9) | | | 3 (8.3) | | |
| Anxiety | | | | | | |
| No | 21 (1.5) | 51.27 | **<0.001** | 13 (0.9) | 38.80 | **<0.001** |
| Yes | 13 (12.3) | | | 9 (8.5) | | |

Fisher's exact test was applied for categorical variables with low expected cell frequencies (<5).
Bold values indicate statistically significant results (*p* < 0.05).

Past-year suicide attempts exhibited similar patterns. Urban adolescents reported suicide attempts, whereas no participants in rural areas did so ($\chi^2 = 9.40$, $p = 0.002$). Grade 8 students showed the highest rates compared to other grades ($\chi^2 = 5.58$, $p = 0.062$). Smoking and alcohol use were also significantly associated with past-year attempts ($p < 0.05$ for both). Adolescents with depression showed higher prevalence than those without ($p = 0.029$; Table 3).

### Multivariable associations between study variables and suicidal ideation

The logistic regression for lifetime suicidal ideation showed a strong model fit, with a Nagelkerke $R^2$ of 38.1% and a nonsignificant Hosmer–Lemeshow test ($\chi^2 = 5.930$, $p = 0.655$), explaining a significant portion of the variance. Adolescents aged 15–17 years were almost twice as likely to report lifetime suicidal ideation compared to those aged 12–14 years (aOR = 1.98, 95% CI: 1.03–3.80, $p = 0.040$). Adolescents living in rural areas were found to have a significantly higher likelihood of lifetime suicidal ideation compared to their urban counterparts (aOR = 6.86, 95% CI: 2.54–18.55, $p < 0.001$). A significant association was also observed for family type, with adolescents from nuclear families being nearly five times more likely to report lifetime suicidal ideation (aOR = 4.93, 95% CI: 1.49–16.28, $p = 0.009$). Additionally, smoking was strongly associated with lifetime suicidal ideation (aOR = 4.14, 95% CI: 1.57–10.90, $p = 0.004$). Depression and anxiety were also significant factors, with adolescents suffering from depression (aOR = 2.35, 95% CI: 1.19–4.62, $p = 0.013$) and anxiety (aOR = 5.13, 95% CI: 2.31–11.41, $p < 0.001$) being more likely to experience lifetime suicidal ideation.

For past-year suicidal ideation, the model had a Nagelkerke $R^2$ of 35%, and the Hosmer–Lemeshow test ($\chi^2 = 11.751$, $p = 0.163$) indicated an acceptable fit. Rural adolescents had 25 times the odds of past-year suicidal ideation compared to their urban adolescents (aOR = 25.17, 95% CI: 2.69–235.54, $p = 0.005$). Smoking was again significantly associated with past-year suicidal ideation (aOR = 5.06, 95% CI: 1.46–17.51, $p = 0.010$). Although not reaching statistical significance, depression (aOR = 2.29, 95% CI: 0.93–5.59, $p = 0.069$) remained a notable factor, while anxiety was a significant factor (aOR = 2.92, 95% CI: 1.04–8.17, $p = 0.041$), suggesting its continued impact on suicidal thoughts within the past year (Table 2).

### Multivariable associations between study variables and suicide plans

The logistic regression model for lifetime suicide plans demonstrated a good fit, with a Nagelkerke $R^2$ of 29.5% and a nonsignificant Hosmer–Lemeshow test ($\chi^2 = 7.652$, $p = 0.468$). Adolescents from joint families were significantly less likely to report lifetime suicide plans compared to those from nuclear families (aOR = 0.11, 95% CI: 0.01–0.94, $p = 0.044$). Higher paternal education levels were associated with a reduced likelihood of suicide plans, with adolescents of fathers who had secondary education showing a significant protective effect (aOR = 0.31, 95% CI: 0.17–0.94, $p = 0.020$). Smoking and anxiety were strong predictors of lifetime suicide plans. Adolescents who smoked had more than six times the odds of reporting suicide plans (aOR = 6.41, 95% CI: 1.83–22.45, $p = 0.004$), and those with anxiety had over six times the odds (aOR = 6.40, 95% CI: 1.92–21.33, $p = 0.002$; Table 5).

For past-year suicide plans, the model also showed good fit (Nagelkerke $R^2 = 31.9\%$, $\chi^2 = 5.290$, $p = 0.726$). While no significant associations were observed for most demographic variables, smoking and anxiety remained strong predictors. Adolescents who smoked had over two and a half times the odds of reporting past-year suicide plans (aOR = 2.66, 95% CI: 0.49–14.23, $p = 0.252$), and those with anxiety had more than seven times the odds (aOR = 7.29, 95% CI: 1.66–31.86, $p = 0.008$). Although not

**Table 3.** Associations between the study variables and suicide plans

| Variables | Lifetime suicidal attempt | | | Past year suicide attempt | | |
|---|---|---|---|---|---|---|
| | Yes, *n* (%) | $\chi^2$ test value | *p*-value | Yes, *n* (%) | $\chi^2$ test value | *p*-value |
| Age group (mean ± SD = 13.91 ± 0.02) | | | | | | |
| 12–14 | 8 (0.7) | 0.17 | 0.674 | 6 (0.6) | 0.03 | 0.854 |
| 15–17 | 4 (1.0) | | | 2 (0.5) | | |
| Gender | | | | | | |
| Boys | 6 (0.7) | 0.07 | 0.783 | 3 (0.4) | 1.54 | 0.213 |
| Girls | 6 (0.9) | | | 6 (0.9) | | |
| Grade | | | | | | |
| 7 | 0 (0.0) | 5.40 | 0.067 | 0 (0) | 5.58 | 0.062 |
| 8 | 8 (1.3) | | | 6 (1) | | |
| 9 | 3 (0.7) | | | 1 (0.2) | | |
| Location | | | | | | |
| Urban | 11 (1.5) | 8.78 | **0.003** | 9 (1.2) | 9.40 | **0.002** |
| Rural | 1 (0.1) | | | 0 (0) | | |
| Number of family members | | | | | | |
| <5 | 9 (0.8) | 0.28 | 0.866 | 6 (0.6) | 0.19 | 0.909 |
| 6–10 | 3 (0.8) | | | 2 (0.5) | | |
| >10 | 0 (0) | | | 0 (0) | | |
| Family type | | | | | | |
| Nuclear | 12 (1) | 2.42 | 0.120 | 8 (0.7) | 0.20 | 0.654 |
| Joint | 0 (0) | | | 1 (0.4) | | |
| Birth order | | | | | | |
| First | 8 (1.1) | 2.91 | 0.233 | 4 (0.5) | 1.87 | 0.392 |
| Second | 3 (0.7) | | | 3 (0.7) | | |
| Third or more | 0 (0) | | | 0 (0) | | |
| BMI | | | | | | |
| Underweight | 2 (0.3) | 2.76 | 0.251 | 1 (0.2) | 3.38 | 0.184 |
| Normal | 8 (1.1) | | | 7 (1) | | |
| Overweight or obese | 2 (1.3) | | | 1 (0.6) | | |
| Father's education | | | | | | |
| Primary or less | 1 (0.2) | 4.41 | 0.110 | 1 (0.2) | 5.69 | 0.058 |
| Secondary | 6 (1.1) | | | 3 (0.6) | | |
| Tertiary | 5 (1.7) | | | 5 (1.7) | | |
| Mother's education | | | | | | |
| Primary or less | 2 (0.5) | 2.60 | 0.273 | 1 (0.2) | 4.62 | 0.099 |
| Secondary | 7 (1) | | | 5 (0.7) | | |
| Tertiary | 3 (1.9) | | | 3 (1.9) | | |
| Father's occupation | | | | | | |
| Employed | 6 (1.6) | 5.96 | 0.051 | 5 (1.4) | 5.12 | 0.077 |
| Business | 5 (1) | | | 3 (0.6) | | |
| Others | 1 (0.2) | | | 1 (0.2) | | |
| Mother's occupation | | | | | | |
| Service | 0 (0) | 1.05 | 0.304 | 0 (0) | 0.86 | 0.352 |
| Housewife | 11 (0.8) | | | 9 (0.7) | | |

(*Continued*)

**Table 3.** (*Continued*)

| Variables | Lifetime suicidal attempt | | | Past year suicide attempt | | |
|---|---|---|---|---|---|---|
| | Yes, *n* (%) | $\chi^2$ test value | *p*-value | Yes, *n* (%) | $\chi^2$ test value | *p*-value |
| Ever had smoking | | | | | | |
| No | 8 (0.6) | 20.67 | **<0.001** | 7 (0.5) | 5.72 | **0.017** |
| Yes | 4 (5.6) | | | 2 (2.8) | | |
| Ever had alcohol | | | | | | |
| No | 11 (0.8) | 6.85 | **0.009** | 8 (0.6) | 9.72 | **0.002** |
| Yes | 1 (7.1) | | | 1 (7.1) | | |
| Depression | | | | | | |
| No | 7 (0.6) | 5.14 | **0.023** | 5 (0.4) | 4.76 | **0.029** |
| Yes | 5 (2.0) | | | 4 (1.6) | | |
| Insomnia | | | | | | |
| No | 11 (0.8) | 1.80 | 0.179 | 8 (0.5) | 2.92 | 0.087 |
| Yes | 1 (2.8) | | | 1 (2.8) | | |
| Anxiety | | | | | | |
| No | 8 (0.6) | 12.65 | **<0.001** | 7 (0.5) | 3.15 | 0.076 |
| Yes | 4 (3.8) | | | 2 (1.9) | | |

Fisher's exact test was applied for categorical variables with low expected cell frequencies (<5).
Bold values indicate statistically significant results (*p* < 0.05).

statistically significant, joint family type showed a protective trend against suicide plans (aOR = 0.18, 95% CI: 0.20–1.64, *p* = 0.129; Table 5).

### Multivariable associations between study variables and suicide attempts

The logistic regression model for lifetime suicide attempts showed a moderate fit, with a Nagelkerke $R^2$ of 20.2% and a nonsignificant Hosmer–Lemeshow test ($\chi^2$ = 6.563, *p* = 0.584). Smoking was a strong predictor of lifetime suicide attempts, with adolescents who smoked having nearly 10 times the odds of reporting attempts (aOR = 9.79, 95% CI: 2.23–42.88, *p* = 0.002). While anxiety approached significance (aOR = 4.83, 95% CI: 0.81–28.77, *p* = 0.083), no other factors showed statistically significant associations (Table 6).

For past-year suicide attempts, the model also demonstrated a moderate fit (Nagelkerke $R^2$ = 19.9%, $\chi^2$ = 1.780, *p* = 0.987). Only alcohol consumption was a significant predictor, with adolescents who consumed alcohol having approximately 23 times the odds of reporting past-year attempts (aOR = 23.19, 95% CI: 1.21–442.17, *p* = 0.037). No significant associations were observed for smoking or anxiety, and demographic variables did not show significant effects on past-year suicide attempts (Table 6).

### Discussion

The present study provides significant insights into the prevalence and associated factors of suicidal behaviors among adolescents in diverse settings. Findings reveal that 6.8% of participants experienced lifetime suicidal ideation, 2.3% made suicide plans and 0.8% attempted suicide. Within the past year, these figures were 3.2%, 1.5% and 0.6%, respectively. Key factors of lifetime suicidal ideation included being aged 15–17 years, residing in rural areas, coming from nuclear families, having a history of smoking and suffering from depression and anxiety. Anxiety emerged as a strong predictor of both lifetime and past-year suicide plans and attempts, alongside smoking, while alcohol use was significantly associated with past-year suicide attempts. These results highlight the need for targeted interventions and policies on mental health support for adolescents, particularly those in rural settings or with high-risk characteristics.

The prevalence of suicidal behaviors among adolescents in this study reveals both consistencies and deviations from prior research, highlighting the complex interplay of cultural, social and environmental factors influencing suicidal behaviors. The lifetime prevalence of suicidal ideation in our study is slightly higher than previous figures from rural Bangladeshi adolescents, which reported a 5% rate (Begum et al. 2017). This variation could stem from differences in sampling settings, with our study encompassing both urban and rural populations, potentially capturing a broader spectrum of stressors and social determinants. Conversely, our rates for suicide plans (2.3%) and attempts (0.8%) are lower than those observed in earlier studies of rural community populations, which reported 5.5% for plans and 1.8% for attempts (Mamun et al. 2022a). Notably, our findings indicate that adolescents from joint families were significantly less likely to report lifetime suicide plans, suggesting that the social support structures inherent in joint family systems may provide protective benefits. These discrepancies may also reflect differences in reporting behaviors in school-based setting or cultural factors that discourage adolescents from disclosing sensitive information. Additionally, the stigma surrounding mental health and suicidal behaviors in Bangladesh might have contributed to the lower reporting of these behaviors in the current study.

Compared to global data, our findings show notably lower rates than those reported in the 2014 Global School-Based Student Health Survey in Bangladesh, which recorded 4.9% for ideation, 7.4% for planning and 6.7% for attempts (Khan et al. 2020). These lower rates could reflect potential underreporting in our sample or a

**Table 4.** Multivariable associations between study variables and suicidal ideation

| Variable name | Lifetime suicide ideation (Nagelkerke $R^2$ = 38.1%; HL test, $\chi^2$ = 5.930, $p$ = 0.655) | | Past year suicide ideation (Nagelkerke $R^2$ = 35%; HL test, $\chi^2$ = 11.751, $p$ = 0.163) | |
|---|---|---|---|---|
| | aOR, 95% CI | $p$-value | aOR, 95% CI | $p$-value |
| **Age group** | | | | |
| 12–14 | Reference | **0.040** | Reference | 0.108 |
| 15–17 | 1.98 (1.03–3.80) | | 2.05 (0.85–4.93) | |
| **Gender** | | | | |
| Boys | Reference | 0.350 | Reference | 0.431 |
| Girls | 1.33 (0.72–2.45) | | 1.38 (0.61–3.11) | |
| **Grade** | | | | |
| 7 | Reference | 0.281 | Reference | 0.338 |
| 8 | 1.97 (0.75–5.12) | | 2.40 (0.64–8.83) | |
| 9 | 2.24 (0.82–6.14) | | 1.61 (0.39–6.58) | |
| **Location** | | | | |
| Urban | Reference | **<0.001** | Reference | **0.005** |
| Rural | 6.86 (2.54–18.55) | | 25.17 (2.69–235.54) | |
| **Number of family members** | | | | |
| <5 | Reference | 0.883 | Reference | 0.489 |
| 6–10 | 1 (0.45–2.20) | | 0.45 (0.12–1.65) | |
| >10 | 1.65 (0.21–12.48) | | 0.73 (0.05–10.70) | |
| **Family type** | | | | |
| Nuclear | 4.93 (1.49–16.28) | **0.009** | 1.49 (0.37–6.07) | 0.571 |
| Joint | Reference | | Reference | |
| **Birth order** | | | | |
| First | 1.07 (0.43–2.66) | 0.226 | 0.85 (0.24–2.99) | 0.315 |
| Second | 0.58 (0.21–1.58) | | 0.41 (0.09–1.74) | |
| Third or more | Reference | | Reference | |
| **Father's education** | | | | |
| Primary or less | 0.62 (0.19–199) | 0.722 | 0.44 (0.07–2.58) | 0.364 |
| Secondary | 0.81 (0.39–1.70) | | 0.49 (0.18–1.34) | |
| Tertiary | Reference | | Reference | |
| **Mother's education** | | | | |
| Primary or less | 0.78 (0.20–2.97) | 0.672 | 0.84 (0.13–5.43) | 0.232 |
| Secondary | 0.68 (0.29–1.61) | | 0.44 (0.16–1.23) | |
| Tertiary | Reference | | Reference | |
| **BMI** | | | | |
| Underweight | 1.07 (0.37–3.09) | 0.320 | 0.82 (0.20–3.29) | 0.388 |
| Normal | 1.65 (0.67–4.09) | | 1.58 (0.52–4.71) | |
| Overweight or more | Reference | | Reference | |
| **Father's occupation** | | | | |
| Employed | 0.68 (0.26–1.81) | 0.174 | 1.04 (0.27–4.01) | 0.595 |
| Business | 1.32 (0.55–3.14) | | 1.54 (0.43–5.44) | |
| Other | Reference | | Reference | |
| **Mother's occupation** | | | | |
| Service | 1.46 (0.59–3.57) | 0.408 | 1.13 (0.38–3.35) | 0.824 |

(*Continued*)

**Table 4.** (*Continued*)

| Variable name | Lifetime suicide ideation (Nagelkerke $R^2$ = 38.1%; HL test, $\chi^2$ = 5.930, p = 0.655) | | Past year suicide ideation (Nagelkerke $R^2$ = 35%; HL test, $\chi^2$ = 11.751, p = 0.163) | |
|---|---|---|---|---|
| | aOR, 95% CI | *p*-value | aOR, 95% CI | *p*-value |
| Housewife | Reference | | Reference | |
| **Ever had smoking** | | | | |
| No | Reference | **0.004** | Reference | **0.010** |
| Yes | 4.14 (1.57–10.90) | | 5.06 (1.46–17.51) | |
| **Ever had alcohol** | | | | |
| No | Reference | 0.321 | Reference | 0.601 |
| Yes | 2.54 (0.40–16.05) | | 1.93 (0.16–22.69) | |
| **Depression** | | | | |
| No | Reference | **0.013** | Reference | 0.069 |
| Yes | 2.35 (1.19–4.62) | | 2.29 (0.93–5.59) | |
| **Insomnia** | | | | |
| No | Reference | 0.402 | Reference | 0.535 |
| Yes | 1.62 (0.52–5.02) | | 0.58 (0.10–3.16) | |
| **Anxiety** | | | | |
| No | Reference | **<0.001** | Reference | **0.041** |
| Yes | 5.13 (2.31–11.41) | | 2.92 (1.04–8.17) | |

Bold values indicate statistically significant results (*p* < 0.05).

more recent shift in societal attitudes toward suicide, possibly due to increased mental health awareness efforts within schools and communities. Global comparisons reveal notable variability in suicidal behaviors across different cultural and regional contexts. For example, adolescents in Namibia exhibited significantly higher rates of suicidal ideation (20.2%), plans (25.2%) and attempts (24.5%; Quarshie et al. 2023). Similarly, higher rates have been reported in Liberia (Quarshie et al. 2020), West African countries [Ghana, Benin, Liberia and Sierra Leone (Diallo et al. 2023), Samoa (Sarfo et al. 2023) and Bhutan (Dema et al. 2019)]. These differences may result from varying cultural norms, variations in suicide literacy, access to mental health resources and differing levels of stigma. For example, adolescents in countries with lower mental health literacy and higher stigma may experience stronger barriers to seeking help, which could influence the prevalence of reported suicidal behaviors. In Bangladesh, lower suicide literacy and a strong relationship with stigmatizing attitudes have been documented among young adults (Jahan et al. 2023), potentially contributing to the lower rates observed in this study.

Previous evidence indicates that individuals in rural areas face a higher risk of suicidal ideation compared to their urban counterparts. For example, a study involving 5,926 participants in China found that the rate of suicidal ideation was 2.8% among rural residents, compared to 1.8% in urban areas (Ma et al. 2009). In rural areas, limited access to mental health resources, social isolation and economic challenges may heighten psychological distress, contributing to these disparities (Margerison and Goldman-Mellor 2019). Similarly, rural women have shown increased odds of lifetime suicidal ideation compared to those from urban areas (Margerison and Goldman-Mellor 2019). Chinese college students from rural regions also reported approximately double the risk of suicidal ideation compared to their urban peers (Meng et al. 2013). These findings are consistent with the results of our study, where

adolescents in rural areas showed elevated rates of suicidal ideation. This indicates the needs for targeted interventions to address the unique challenges faced by rural communities, aiming to reduce elevated risk of suicidal behaviors.

Adolescents often engage in exploratory behaviors like smoking which significantly influence mental health and potentially escalate to more severe outcomes, such as suicidal behaviors. Our study aligns with previous research linking smoking to suicidal ideation and suggests that smoking may serve as both a coping mechanism and a risk factor. For example, a comprehensive meta-analysis encompassing 63 studies demonstrated that current smoking doubles the risk of suicidal ideation (Poorolajal and Darvishi 2016). Similar associations have been observed among college students (Waters et al. 2021) and Brazilian adolescents (Slomp et al. 2019). Differences in smoking prevalence and attitudes toward smoking across cultural contexts could contribute to varying rates of smoking-associated suicidal ideation in adolescent populations (Page and Danielson 2011; Poorolajal and Darvishi 2016). In Bangladesh, smoking is often socially discouraged, particularly among younger individuals, which might contribute to lower reported rates (Saha et al. 2024). However, adolescents who do smoke may experience heightened psychological distress, increasing their vulnerability to suicidal ideation (Slomp et al. 2019). This suggests that implementing targeted interventions to promote smoking cessation among adolescents could be crucial in mitigating the risk of suicidal behaviors.

Mental health issues, including psychological disorders such as depression and anxiety, are strongly associated with suicidal behaviors. These conditions may exacerbate negative thought patterns and emotional distress among adolescents, thereby increasing the risk of suicidal behaviors. This finding is consistent with previous research involving sub-Saharan African adolescents (Nyundo et al. 2020), Nepali youth (Pandey et al. 2019) and US university students

**Table 5.** Multivariable associations between the selected study variables and suicide plans

| Variable name | Lifetime suicide plan (Nagelkerke $R^2$ = 29.5%; HL test, $\chi^2$ = 7.652, p = 0.468) | | Past year suicide plan (Nagelkerke $R^2$ = 31.9%; HL test, $\chi^2$ = 5.290, p = 0.726) | |
|---|---|---|---|---|
| | aOR, 95% CI | *p*-value | aOR, 95% CI | *p*-value |
| Age group | | | | |
| 12–14 | Reference | 0.815 | Reference | 0.660 |
| 15–17 | 0.88 (0.30–2.54) | | 0.73 (0.18–2.96) | |
| Gender | | | | |
| Boys | Reference | 0.795 | Reference | 0.746 |
| Girls | 0.88 (0.33–2.30) | | 0.74 (0.22–2.53) | |
| Family type | | | | |
| Nuclear | Reference | **0.044** | Reference | 0.129 |
| Joint | 0.11 (0.01–0.94) | | 0.18 (0.20–1.64) | |
| Birth order | | | | |
| First | Reference | 0.593 | Reference | 0.740 |
| Second | 0.61 (0.21–1.77) | | 0.85 (0.23–3.06) | |
| Third or more | 0.58 (0.11–2.90) | | 0.41 (0.04–3.89) | |
| Father's education | | | | |
| Primary or less | 0.41 (.003–0.50) | **0.020** | Reference | 0.192 |
| Secondary | 0.31 (0.17–0.94) | | 0.86 (0.07–9.49) | |
| Tertiary | Reference | | 3.84 (0.31–47.70) | |
| Mother's education | | | | |
| Primary or less | Reference | 0.956 | Reference | 0.379 |
| Secondary | 0.87 (0.14–5.46) | | 0.56 (0.05–6.13) | |
| Tertiary | 1.03 (0.12–8.80) | | 1.58 (0.10–24.64) | |
| BMI | | | | |
| Underweight | Reference | 0.847 | Reference | 0.474 |
| Normal | 1.20 (0.40–3.55) | | 2.72 (0.54–13.64) | |
| Overweight or more | 1.53 (0.35–6.64) | | 2.44 (0.28–20.87) | |
| Father's occupation | | | | |
| Employed | Reference | 0.351 | Reference | 0.659 |
| Business | 1.89 (0.70–5.09) | | 1.62 (0.46–5.670 | |
| Other | 0.90 (0.19–4.12) | | 0.86 (0.12–5.90) | |
| Mother's occupation | | | | |
| Service | Reference | 0.977 | Reference | 0.784 |
| Housewife | 0.98 (0.24–3.95) | | 0.80 (0.17–3.73) | |
| Ever had smoking | | | | |
| No | Reference | **0.004** | Reference | 0.252 |
| Yes | 6.41 (1.83–22.45) | | 2.66 (0.49–14.23) | |
| Ever had alcohol | | | | |
| No | Reference | 0.115 | Reference | 0.184 |
| Yes | 5.11 (0.67–38.90) | | 6.37 (0.41–97.72) | |
| Depression | | | | |
| No | Reference | 0.533 | Reference | 0.249 |
| Yes | 1.44 (0.45–4.53) | | 2.33 (0.55–9.81) | |

*(Continued)*

**Table 5.** (*Continued*)

| Variable name | Lifetime suicide plan (Nagelkerke $R^2$ = 29.5%; HL test, $\chi^2$ = 7.652, *p* = 0.468) | | Past year suicide plan (Nagelkerke $R^2$ = 31.9%; HL test, $\chi^2$ = 5.290, *p* = 0.726) | |
|---|---|---|---|---|
| | aOR, 95% CI | *p*-value | aOR, 95% CI | *p*-value |
| Insomnia | | | | |
| No | Reference | 0.452 | Reference | 0.859 |
| Yes | 1.80 (0.38–8.46) | | 1.19 (0.17–8.28) | |
| Anxiety | | | | |
| No | Reference | **0.002** | Reference | **0.008** |
| Yes | 6.40 (1.92–21.33) | | 7.29 (1.66–31.86) | |

Bold values indicate statistically significant results (*p* < 0.05).

**Table 6.** Multivariable associations between the selected study variables and suicide attempts

| Variable name | Lifetime suicidal attempt (Nagelkerke $R^2$ = 20.2%; HL test, $\chi^2$ = 6.563, *p* = 0.584) | | Past-year suicide attempt (Nagelkerke $R^2$ = 19.9%; HL test, $\chi^2$ = 1.780, *p* = 0.987) | |
|---|---|---|---|---|
| | aOR, 95% CI | *p*-value | aOR, 95% CI | *p*-value |
| Age group | | | | |
| 12–14 | Reference | 0.972 | Reference | 0.953 |
| 15–17 | 1.02 (0.25–4.07) | | 0.94 (0.16–5.42) | |
| Gender | | | | |
| Boys | Reference | 0.892 | Reference | 0.503 |
| Girls | 1.09 (0.29–4.10) | | 1.73 (0.34–8.76) | |
| Father's education | | | | |
| Primary or less | 0.23 (0.01–4.25) | 0.542 | 1.76 (0.07–41.92) | 0.892 |
| Secondary | 0.98 (0.22–4.41) | | 0.12–6.17) | |
| Tertiary | Reference | | Reference | |
| Mother's education | | | | |
| Primary or less | Reference | 0.855 | Reference | 0.687 |
| Secondary | 0.58 (0.07–4.26) | | 0.81 (0.05–12.24) | |
| Tertiary | 0.69 (0.06–8.12) | | 1.82 (0.08–40.82) | |
| BMI | | | | |
| Underweight | Reference | 0.574 | Reference | 0.667 |
| Normal | 2.31 (0.44–11.94) | | 2.73 (0.30–24.82) | |
| Overweight or more | 2.63 (0.31–22.32) | | 2.19 (0.12–40.20) | |
| Father's occupation | | | | |
| Employed | Reference | 0.393 | Reference | 0.527 |
| Business | 0.66 (0.18–2.41) | | 0.38 (0.07–1.99) | |
| Other | 0.19 (0.01–2.09) | | – | |
| Ever had smoking | | | | |
| No | Reference | **0.002** | Reference | 0.720 |
| Yes | 9.79 (2.23–42.88) | | 1.60 (0.12–21.51) | |
| Ever had alcohol | | | | |
| No | Reference | 0.265 | Reference | **0.037** |
| Yes | 4.29 (0.33–42.88) | | 23.19 (1.21–442.17) | |
| Depression | | | | |
| No | Reference | 0.855 | Reference | 0.943 |
| Yes | 0.85 (0.15–4.74) | | 0.92 (0.11–7.21) | |

(*Continued*)

**Table 6.** (*Continued*)

| Variable name | Lifetime suicidal attempt (Nagelkerke $R^2$ = 20.2%; HL test, $\chi^2$ = 6.563, *p* = 0.584) | | Past-year suicide attempt (Nagelkerke $R^2$ = 19.9%; HL test, $\chi^2$ = 1.780, *p* = 0.987) | |
|---|---|---|---|---|
| | aOR, 95% CI | *p*-value | aOR, 95% CI | *p*-value |
| Insomnia | | | | |
| No | Reference | 0.777 | Reference | 0.374 |
| Yes | 1.40 (0.13–14.66) | | 3.13 (0.25–38.98) | |
| Anxiety | | | | |
| No | Reference | 0.083 | Reference | 0.460 |
| Yes | 4.83 (0.81–28.77) | | 2.34 (0.24–22.55) | |

Bold values indicate statistically significant results ( *p* < 0.05).

(Becker et al. 2018). Similar patterns have been observed in Bangladesh, where studies on rural populations (Mamun et al. 2022a), university entrance exam candidates (Mamun et al. 2022b; Nahrin et al. 2023) and general adolescents (Khan et al. 2020) highlighted the impact of mental health problems on suicidal ideation. The observed consistency may be attributed to universal factors, such as the physiological effects of depression and anxiety on cognition and decision-making, which transcend cultural differences. Addressing these mental health challenges is critical, as early intervention can reduce the risk of escalating to suicidal behaviors. It is crucial for stakeholders to create supportive environments and provide accessible mental health resources tailored to adolescents' needs, particularly for those at higher risk due to these mental health challenges.

The current study presents several notable strengths that significantly contribute to the existing literature on adolescent mental health and suicidal behaviors. The use of a two-stage sampling technique enhances the robustness of the methodology, providing valuable insights into both lifetime and past-year suicidal behaviors among adolescents across diverse settings. This approach not only improves the generalizability of the findings but also offers a comprehensive view of the prevalence and associated factors of suicidal tendencies within this population. Additionally, the large sample size increases the statistical power of the study, allowing for more precise estimates.

However, the study's cross-sectional design limits its ability to establish causal relationships between the factors examined and suicidal behaviors among adolescents. Future research employing longitudinal designs would be beneficial to explore how adolescent experiences and exposures impact outcomes in adulthood. Additionally, the study's sample size was drawn from a single district and included only school-going adolescents, which may affect the generalizability of the findings to other regions and adolescents in religious or vocational institutions. Future studies should aim to include more diverse samples across multiple districts and institutional types to capture the broader adolescent population in Bangladesh. Another limitation is the study's focus on selected socio-demographic and behavioral factors, which did not account for impulsivity, a known factor closely linked to suicidal behaviors, particularly in youth. Impulsivity can play a significant role in impulsive suicide attempts among adolescents, and future research should incorporate assessments of impulsivity to provide a more comprehensive understanding of its impact on suicidal behavior in this age group (Moore et al. 2023). While binary measures of lifestyle factors, such as smoking and alcohol use, offered initial insights into substance use among adolescents, they did not capture detailed patterns or underlying reasons for substance use, which could further contextualize their relationship with suicidal behaviors. The cross-sectional nature of the data also prevents establishing causation between these lifestyle factors and suicidal outcomes. Moreover, incorporating additional variables, such as specific school and family-related factors, would further enrich our understanding of the complex elements contributing to adolescent mental health and well-being. These additions could provide more comprehensive insights into the socio-environmental context influencing suicidal behaviors in Bangladeshi adolescents.

## Conclusion

In conclusion, this study provides valuable insights into the prevalence and associated factors of suicidal behaviors among adolescents, highlighting significant associations with demographic, psychosocial and mental health factors. The findings highlight the critical need for targeted interventions that address the unique challenges faced by adolescents, particularly those in rural areas, those engaging in smoking and those experiencing depression and anxiety. While the study's robust methodology and large sample size enhance the reliability of the results, the cross-sectional design limits the ability to infer causality. Future research should focus on longitudinal approaches and explore additional variables to deepen the understanding of the factors influencing adolescent mental health. By addressing these areas, stakeholders can better develop strategies to mitigate suicidal behaviors and promote overall well-being among adolescents.

**Open peer review.** To view the open peer review materials for this article, please visit http://doi.org/10.1017/gmh.2025.10.

**Supplementary material.** The supplementary material for this article can be found at http://doi.org/10.1017/gmh.2025.10.

**Data availability statement.** The datasets will be made available to appropriate academic parties upon request from the corresponding author.

**Acknowledgements.** The authors are grateful to all the participants and team members who contributed to the project. In addition, we would like to express our gratitude to the school staff and the CSF Global staff for their support during the study implementation.

**Author contribution.** This study was conceptualized by FAM and MAM, with support from the other authors. FAM and MAM led the implementation and management of the project, while JI and MM provided logistical support. Data analysis and the initial draft of the manuscript were carried out by FAM.

Significant contributions were made by AAH, JI, MMA, MAM and MM at various stages, including data management, analysis and manuscript revision. All authors engaged in extensive editing and review of the manuscript. The final version was approved by all authors.

**Financial support.** This study was funded by the University of South Asia, Dhaka, Bangladesh. Besides, Dr. ALmerab has to acknowledge that the funding support from currently receiving as Princess Nourah bint Abdulrahman University Researchers Supporting Project number (PNURSP2025R563), Princess Nourah bint Abdulrahman University, Riyadh, Saudi Arabia.

**Competing interest.** The authors of the research work do not have any conflict of interest.

**Ethics statement.** The study was conducted in accordance with the ethical principles of the Helsinki Declaration, 2013. Ethical approval for the study was granted by [institution name was blinded for review] after evaluation of the research proposal. Written informed consent was obtained from school authorities, parents and all participants, ensuring they were fully aware of the study's purpose, procedures and their right to withdraw at any time. To ensure participant confidentiality, all data were anonymized and securely stored throughout the study. Moreover, we implemented a protocol allowing participants to withdraw from the study if they felt distressed, with immediate support available.

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
