## [Reviewer Report]

o Summary of the paper

□ This study investigates the prevalence and factors associated with suicidal behaviors among adolescents in Bangladesh, a population for which limited data is available. Participants from both urban and rural areas were recruited using a two-stage stratified cluster sampling approach. The study gathered data on socio-demographic factors, smoking, alcohol use, depression, anxiety, and insomnia. Lifetime and 12-month prevalence rates of suicidal behaviors (i.e., ideation, planning, and attempts) were assessed, while logistic regression identified factors related to suicidal ideation only. The findings showed that 6.8% of participants reported lifetime suicidal ideation, with 2.3% having made suicide plans and 0.8% attempting suicide. Over the past 12 months, the prevalence rates were 3.2%, 1.5%, and 0.6%, respectively. Key factors linked to suicidal ideation included being aged 15-17 years, living in rural areas, belonging to nuclear families, smoking, depression, and anxiety. The article contributes to the limited suicide literature in Bangladesh, underscoring the need for targeted interventions and policies to address adolescent mental health and suicidal behaviors.

o Key Points

□ The study contributes valuable knowledge to the sparse literature on suicide in low- and middle-income countries (LMICs).

□ The work is highly significant and fills a gap in the literature. It adds important insights into a vulnerable population that has not been widely studied.

□ The analysis is thorough and well-executed. However, it would be stronger if the same depth of analysis applied to suicidal ideation was extended to suicidal planning and attempts (see major concerns below).

o Major issues

□ Focus on Suicidal Ideation Only: A key concern is why the authors focused solely on the association between variables and suicidal ideation, excluding suicidal plans and attempts from the analysis. Since the prevalence rates for all three outcomes are presented, a more comprehensive analysis would have enriched the paper. If there was a compelling reason for this omission, it should be clearly stated in the paper. Otherwise, including the additional analyses would make the study more robust, especially given that the paper is framed as addressing “suicidal behaviors” rather than just “suicidal ideation.”

o Minor issues

• Abstract Clarity:

• The abstract should clarify that univariate analyses were conducted using chi-square and Fisher’s exact tests, followed by multivariable logistic regression to identify factors associated with suicidal behaviors. The phrasing is currently a bit unclear: “ Statistical analyses, including Chi-square and Fisher’s exact test, were used to determine the association between suicidal ideation and the study variables.”

• Regarding the 12-month suicidal behavior rates, the phrase “rates decreased slightly” could be misleading, as this implies a longitudinal study. Instead, the authors should simply state the 12-month prevalence rates in this context; for example, “The 12-month prevalence rates were….”.

• The purpose of the study should be more explicit. Is the focus on assessing prevalence rates, risk factors, or both? For which outcomes? Making this clear in the abstract would improve its clarity.

• Phrasing and Language:

• The terms “committed suicide” and “committing suicide” used in lines 49 and 135 are outdated and stigmatizing. More appropriate language would be “died by suicide” or “engaged in suicidal behavior.”

• Introduction:

• A stronger rationale should be provided for how this risk factor analysis differs from previous studies, especially since the risk factor analysis focuses on suicidal ideation ONLY. The inclusion of commonly overlooked but important factors is a strength, and so it would be helpful to differentiate this analysis more clearly from past work, as I believe this study includes very important and culturally relevant factors.

• Methods:

• As noted in the major concerns, the rationale for not analyzing risk factors for suicidal plans and attempts should at least be addressed if the analyses will not be conducted.

• The authors should also explain how missing data were handled in their analysis for transparency.

• Consider including an appendix listing all study questions and responses for key measures, or list them in the methods similar to how the mental health items are described.

• Results:

• Consider renaming section 3.3 to “Univariate Associations Between Study Variables and Suicidal Ideation” and section 3.4 to “Multivariable Associations Between Study Variables and Suicidal Ideation” to better highlight the distinction between the two sections.

• Given that this is a cross-sectional study, it would be more accurate to refer to variables as “factors” rather than “predictors,” as the latter implies a longitudinal component.

• Discussion:

• Similar to the abstract, in line 232, avoid saying that the prevalence rates “decreased” since the study is not longitudinal. Instead, report the 12-month prevalence rates without implying a reduction over time.

• The authors should acknowledge the limitations that the inclusion criteria (i.e., being present for the survey) might have excluded adolescents who may have died by suicide. This could affect the generalizability of the findings, and impulsivity, particularly among youth, should also be discussed as a potential limitation due to its particular link to suicide in this age group (e.g., future research should assess impulsive suicide attempts, as this was not evaluated in the current study).

• Grammar and Style:

• The word “Besides” is used frequently as a transition word but does not fit well in many cases. Replacing it with alternatives like “additionally,” “in addition,” or “furthermore” would improve readability.

---

## [Reviewer Report]

Overall comments:

The study’s objective was to investigate suicidal behaviors among adolescents in Bangladesh, focusing on calculating both the prevalence and on exploring determinants of suicidal behaviors within this population. The cross-sectional design, combined with a large sample size, enhances the reliability of the outcome of this study.

However, I have suggestions for the authors to further improve the manuscript’s quality. While it shows potential for publication, it is not ready in its current form and requires extensive revision. Specifically, the in-text citation style needs to be addressed (a comma is missing before the year in some citations). Below are my specific comments for revision:

Impact statement: In 21 line, what kinds of detailed analysis for suicidal behaviors you have done in this study needs to be mentioned here. In 23, why adolescents will be vulnerable population? In 24, how will school administrators and community leaders develop effective mental health programs for adolescents? They are not experts in the mental health field. In 28-suicidal behaviors include suicidal ideation, so no need to include suicidal ideation.

Introduction: Age range for adolescents in Bangladesh is missing. Suicidality study misleads the word by suicidal behavior; hence, I am requesting you to define suicidal behavior. For your reference, you can use the definition from: (1) Self-directed Violence Surveillance: Uniform Definitions and Recommended Data Elements (Version 1.). Atlanta (GA): Centers for Disease Control and Prevention, National Center for Injury Prevention and Control by Crosby, A. E., Ortega, L., & Melanson, C. (2011), and (2) WHO (2014). Here is the document link: https://apps.who.int/iris/bitstream/handle/10665/131056/9789241564779_eng.pdf?sequence=1

In 71-72, you wrote significant factors identified included age and so on. It is not clear what these significant factors are related to. In 75-80, the prevalence of suicidal behaviors was 11.7% was informed by a recent study. Could you please justify why do you need to conduct the same type of research to calculate prevalence and find factors for adolescents?

In 79-Please check the reference, and include parental homework check instead of supervision, because supervision word does not reflect the finding of your cited article. Moreover, you need to find synonyms for rare. Infrequent and rare are not the same in terms of frequencies.

Method: For 2.1 section, you recruited respondents from one district out of 64 districts in the country where the population were school going students. What about other adolescents who did not attend schools or attend religious or vocational curriculum based academic institutions? How do you justify that the sample will represent the entire adolescent population of the country since the prevalence of suicidal behavior varies region to region in Bangladesh? It is important to know because you are capturing the diverse influencing adolescent suicidal behaviors across in Bangladesh, not only in one district.

In 97, it is not clear how you recruited students from randomly selected grades 7, 8, and 9. I understand that these 7, 8, and 9 grades were randomly selected. But how samples were selected from these grades is not clear. Could you please explain it in detail? In addition, in 99, you mention that representation of different age groups, could you please mention the age groups here?

In 100-106, you have elucidated the data collection procedures. I have not found whether the guardian/parents were informed or not, since participants were adolescents. Only permission from school authorities is not enough to collect data from minor aged participants. It is not also clear how research team administered the scale and recorded the data; for example, it is not specified whether the data collection was done in one-on-one sessions or group settings, nor how they addressed risk factors (e.g., psychological well-being if participants experience any distress during providing response against any items of a scale). There is also no clarity on how confidentiality was maintained throughout the process. Furthermore, the composition of the research team remains ambiguous. Were the authors themselves involved in data collection, or were research assistants employed for this task? If research assistants were used, it would be important to know if they received adequate training to ensure the ethical and accurate administration of the research."

It is better to move and merge 107-111 with sample descriptions section (92-99). Based on the inclusion criteria, it appears that random cluster sampling was not utilized. If one of the inclusion criteria is that a sample must be present during the survey period, how did you apply the random sampling technique to recruit participants? Please explain it after 99 line. At the same time, please provide justification that your recruited sample size will be the representative of the population.

In 116-117, you mentioned smoking and alcohol consumption as lifestyle factors and yes/no was the response as measures. In the analysis, you have claimed smoking and alcohol consumption as determinants for suicidal behavior. How these two factors used in your study will be the factors/determinants for suicidal behavior for adolescent population? Yes/No response against statements like ever had alcohol or smoking does not make a causation or relationship between suicidal behavior and these factors. Could you please provide justification with proper citation that smoking, and alcohol consumption can be determinants for suicidal behavior? An example will help to clarify it more precisely. For example, after relationship conflict or breakup with intimate partners, an individual can be addicted to smoking or alcohol. In this case, if that person commits suicide or plan to commit suicide, smoking cannot be the determinant for suicidal behavior, relationship breakup or conflict in relationship will be the determinants.

For 2.2.2 Mental health problems, you have selected depression, insomnia, and anxiety as determinants for suicidal behavior. Could you please explain why you select these three psychological problems?

For 119-133, could you please help me to clarify the following concerns?

1. PHQ was developed for the adult population. Is it reliable to apply PHQ-9 for adolescent population? The language of this scale is English. Have you validated this scale for Bangladeshi people? What was the administration procedures you followed in this study- was it self-administered or administered by research team? Same questions are applicable for ISI-2 and GAD-7 scale.

2. Generalized Anxiety Disorder (GAD) and anxiety does not convey same meaning in the field of mental health (e.g., clinical psychology/psychiatry). Anxiety of GAD is different from anxiety; hence, GAD-7 does not measure anxiety. I am keenly interested in knowing your thoughts against my opinion.

For 2.2.3 section, suicidal thoughts title is not appropriate because it does not include suicidal attempts and suicide by itself. Suicidal behavior or history of suicidal behavior are two proposed appropriate terms. You can accept or propose other appropriate terms. Here, I have an important query to you. Despite having suicidal behavior measuring scales, you have recorded a binary response. Can I know the reason for it?

For 2.3 section, please include your safeguarding protocol in case any respondents feel distress during data collection. It is needed because you collected sensitive data (e.g., mental health problems) that could recall their distressed memories. Please also include the benefits for the research participants. It is better to merge this 2.3 section with data collection procedures because you have a dedicated section for this in 329.

In 151, it is clear that Microsoft Excel 2021 was used to analyze the data, but in 158 you mentioned that you used SPSS version 25 to analyze the data. Did you employ both to analyze the data?

Results: In 163-164, You wrote that sample was stratified into three age groups, but you declared only two age groups which were 12 to 14 years and 15 to 17 years. Please include the third age group. In 165, the percentage of three grades (7, 8, 9) are not equal, it is not a concern for me. But a clarification statement helps readers to understand the concern.

For 3.3 and 3.4 sections, the analysis was for suicidal ideation only where the research title includes suicidal behavior term. I have already shared the resources that articulate the definition of suicidal behavior. Hope you have understood my concern.

I am requesting you to modify the analysis based on the comments against the introduction and methodology section.

Discussion: In 236, please explain how these findings can help to develop the evidence-based interventions and policies?

From 238-248, I have found that several similar research was conducted in the same country and findings were almost same whereas three research were conducted in 2022 and 2020 which is very recent. In this scenario, what is the novelty and what gap this research covers except understanding the prevalence? Please explain it under the problem statement paragraph.

Overall, the discussion section does not reflect the discussion at all except comparing the findings with previous studies. Please articulate your findings through an insightful discussion.

---

## [Editor Report]

The reviewers consider the paper interesting, especially since it is a problem that is rarely studied in countries like Bangladesh, but to be accepted it requires major changes. The authors need to revise the wording as well as the analyses presented. This will require major editing, but we hope the authors will be willing to do so.

Specific comments from the reviewers are detailed below, for the authors to review step by step.

Reviewer 1:

o Key Points

□ The study contributes valuable knowledge to the sparse literature on suicide in low- and middle-income countries (LMICs).

□ The work is highly significant and fills a gap in the literature. It adds important insights into a vulnerable population that has not been widely studied.

□ The analysis is thorough and well-executed. However, it would be stronger if the same depth of analysis applied to suicidal ideation was extended to suicidal planning and attempts (see major concerns below).

o Major issues

□ Focus on Suicidal Ideation Only: A key concern is why the authors focused solely on the association between variables and suicidal ideation, excluding suicidal plans and attempts from the analysis. Since the prevalence rates for all three outcomes are presented, a more comprehensive analysis would have enriched the paper. If there was a compelling reason for this omission, it should be clearly stated in the paper. Otherwise, including the additional analyses would make the study more robust, especially given that the paper is framed as addressing “suicidal behaviors” rather than just “suicidal ideation.”

o Minor issues

• Abstract Clarity:

• The abstract should clarify that univariate analyses were conducted using chi-square and Fisher’s exact tests, followed by multivariable logistic regression to identify factors associated with suicidal behaviors. The phrasing is currently a bit unclear: “ Statistical analyses, including Chi-square and Fisher’s exact test, were used to determine the association between suicidal ideation and the study variables.”

• Regarding the 12-month suicidal behavior rates, the phrase “rates decreased slightly” could be misleading, as this implies a longitudinal study. Instead, the authors should simply state the 12-month prevalence rates in this context; for example, “The 12-month prevalence rates were….”.

• The purpose of the study should be more explicit. Is the focus on assessing prevalence rates, risk factors, or both? For which outcomes? Making this clear in the abstract would improve its clarity.

• Phrasing and Language:

• The terms “committed suicide” and “committing suicide” used in lines 49 and 135 are outdated and stigmatizing. More appropriate language would be “died by suicide” or “engaged in suicidal behavior.”

• Introduction:

• A stronger rationale should be provided for how this risk factor analysis differs from previous studies, especially since the risk factor analysis focuses on suicidal ideation ONLY. The inclusion of commonly overlooked but important factors is a strength, and so it would be helpful to differentiate this analysis more clearly from past work, as I believe this study includes very important and culturally relevant factors.

• Methods:

• As noted in the major concerns, the rationale for not analyzing risk factors for suicidal plans and attempts should at least be addressed if the analyses will not be conducted.

• The authors should also explain how missing data were handled in their analysis for transparency.

• Consider including an appendix listing all study questions and responses for key measures, or list them in the methods similar to how the mental health items are described.

• Results:

• Consider renaming section 3.3 to “Univariate Associations Between Study Variables and Suicidal Ideation” and section 3.4 to “Multivariable Associations Between Study Variables and Suicidal Ideation” to better highlight the distinction between the two sections.

• Given that this is a cross-sectional study, it would be more accurate to refer to variables as “factors” rather than “predictors,” as the latter implies a longitudinal component.

• Discussion:

• Similar to the abstract, in line 232, avoid saying that the prevalence rates “decreased” since the study is not longitudinal. Instead, report the 12-month prevalence rates without implying a reduction over time.

• The authors should acknowledge the limitations that the inclusion criteria (i.e., being present for the survey) might have excluded adolescents who may have died by suicide. This could affect the generalizability of the findings, and impulsivity, particularly among youth, should also be discussed as a potential limitation due to its particular link to suicide in this age group (e.g., future research should assess impulsive suicide attempts, as this was not evaluated in the current study).

• Grammar and Style:

• The word “Besides” is used frequently as a transition word but does not fit well in many cases. Replacing it with alternatives like “additionally,” “in addition,” or “furthermore” would improve readability.

Reviewer 2:

Overall comments:

The study’s objective was to investigate suicidal behaviors among adolescents in Bangladesh, focusing on calculating both the prevalence and on exploring determinants of suicidal behaviors within this population. The cross-sectional design, combined with a large sample size, enhances the reliability of the outcome of this study.

However, I have suggestions for the authors to further improve the manuscript’s quality. While it shows potential for publication, it is not ready in its current form and requires extensive revision. Specifically, the in-text citation style needs to be addressed (a comma is missing before the year in some citations). Below are my specific comments for revision:

Impact statement: In 21 line, what kinds of detailed analysis for suicidal behaviors you have done in this study needs to be mentioned here. In 23, why adolescents will be vulnerable population? In 24, how will school administrators and community leaders develop effective mental health programs for adolescents? They are not experts in the mental health field. In 28-suicidal behaviors include suicidal ideation, so no need to include suicidal ideation.

Introduction: Age range for adolescents in Bangladesh is missing. Suicidality study misleads the word by suicidal behavior; hence, I am requesting you to define suicidal behavior. For your reference, you can use the definition from: (1) Self-directed Violence Surveillance: Uniform Definitions and Recommended Data Elements (Version 1.). Atlanta (GA): Centers for Disease Control and Prevention, National Center for Injury Prevention and Control by Crosby, A. E., Ortega, L., & Melanson, C. (2011), and (2) WHO (2014). Here is the document link: https://apps.who.int/iris/bitstream/handle/10665/131056/9789241564779_eng.pdf?sequence=1

In 71-72, you wrote significant factors identified included age and so on. It is not clear what these significant factors are related to. In 75-80, the prevalence of suicidal behaviors was 11.7% was informed by a recent study. Could you please justify why do you need to conduct the same type of research to calculate prevalence and find factors for adolescents?

In 79-Please check the reference, and include parental homework check instead of supervision, because supervision word does not reflect the finding of your cited article. Moreover, you need to find synonyms for rare. Infrequent and rare are not the same in terms of frequencies.

Method: For 2.1 section, you recruited respondents from one district out of 64 districts in the country where the population were school going students. What about other adolescents who did not attend schools or attend religious or vocational curriculum based academic institutions? How do you justify that the sample will represent the entire adolescent population of the country since the prevalence of suicidal behavior varies region to region in Bangladesh? It is important to know because you are capturing the diverse influencing adolescent suicidal behaviors across in Bangladesh, not only in one district.

In 97, it is not clear how you recruited students from randomly selected grades 7, 8, and 9. I understand that these 7, 8, and 9 grades were randomly selected. But how samples were selected from these grades is not clear. Could you please explain it in detail? In addition, in 99, you mention that representation of different age groups, could you please mention the age groups here?

In 100-106, you have elucidated the data collection procedures. I have not found whether the guardian/parents were informed or not, since participants were adolescents. Only permission from school authorities is not enough to collect data from minor aged participants. It is not also clear how research team administered the scale and recorded the data; for example, it is not specified whether the data collection was done in one-on-one sessions or group settings, nor how they addressed risk factors (e.g., psychological well-being if participants experience any distress during providing response against any items of a scale). There is also no clarity on how confidentiality was maintained throughout the process. Furthermore, the composition of the research team remains ambiguous. Were the authors themselves involved in data collection, or were research assistants employed for this task? If research assistants were used, it would be important to know if they received adequate training to ensure the ethical and accurate administration of the research."

It is better to move and merge 107-111 with sample descriptions section (92-99). Based on the inclusion criteria, it appears that random cluster sampling was not utilized. If one of the inclusion criteria is that a sample must be present during the survey period, how did you apply the random sampling technique to recruit participants? Please explain it after 99 line. At the same time, please provide justification that your recruited sample size will be the representative of the population.

In 116-117, you mentioned smoking and alcohol consumption as lifestyle factors and yes/no was the response as measures. In the analysis, you have claimed smoking and alcohol consumption as determinants for suicidal behavior. How these two factors used in your study will be the factors/determinants for suicidal behavior for adolescent population? Yes/No response against statements like ever had alcohol or smoking does not make a causation or relationship between suicidal behavior and these factors. Could you please provide justification with proper citation that smoking, and alcohol consumption can be determinants for suicidal behavior? An example will help to clarify it more precisely. For example, after relationship conflict or breakup with intimate partners, an individual can be addicted to smoking or alcohol. In this case, if that person commits suicide or plan to commit suicide, smoking cannot be the determinant for suicidal behavior, relationship breakup or conflict in relationship will be the determinants.

For 2.2.2 Mental health problems, you have selected depression, insomnia, and anxiety as determinants for suicidal behavior. Could you please explain why you select these three psychological problems?

For 119-133, could you please help me to clarify the following concerns?

1. PHQ was developed for the adult population. Is it reliable to apply PHQ-9 for adolescent population? The language of this scale is English. Have you validated this scale for Bangladeshi people? What was the administration procedures you followed in this study- was it self-administered or administered by research team? Same questions are applicable for ISI-2 and GAD-7 scale.

2. Generalized Anxiety Disorder (GAD) and anxiety does not convey same meaning in the field of mental health (e.g., clinical psychology/psychiatry). Anxiety of GAD is different from anxiety; hence, GAD-7 does not measure anxiety. I am keenly interested in knowing your thoughts against my opinion.

For 2.2.3 section, suicidal thoughts title is not appropriate because it does not include suicidal attempts and suicide by itself. Suicidal behavior or history of suicidal behavior are two proposed appropriate terms. You can accept or propose other appropriate terms. Here, I have an important query to you. Despite having suicidal behavior measuring scales, you have recorded a binary response. Can I know the reason for it?

For 2.3 section, please include your safeguarding protocol in case any respondents feel distress during data collection. It is needed because you collected sensitive data (e.g., mental health problems) that could recall their distressed memories. Please also include the benefits for the research participants. It is better to merge this 2.3 section with data collection procedures because you have a dedicated section for this in 329.

In 151, it is clear that Microsoft Excel 2021 was used to analyze the data, but in 158 you mentioned that you used SPSS version 25 to analyze the data. Did you employ both to analyze the data?

Results: In 163-164, You wrote that sample was stratified into three age groups, but you declared only two age groups which were 12 to 14 years and 15 to 17 years. Please include the third age group. In 165, the percentage of three grades (7, 8, 9) are not equal, it is not a concern for me. But a clarification statement helps readers to understand the concern.

For 3.3 and 3.4 sections, the analysis was for suicidal ideation only where the research title includes suicidal behavior term. I have already shared the resources that articulate the definition of suicidal behavior. Hope you have understood my concern.

I am requesting you to modify the analysis based on the comments against the introduction and methodology section.

Discussion: In 236, please explain how these findings can help to develop the evidence-based interventions and policies?

From 238-248, I have found that several similar research was conducted in the same country and findings were almost same whereas three research were conducted in 2022 and 2020 which is very recent. In this scenario, what is the novelty and what gap this research covers except understanding the prevalence? Please explain it under the problem statement paragraph.

Overall, the discussion section does not reflect the discussion at all except comparing the findings with previous studies. Please articulate your findings through an insightful discussion.

---

## [Reviewer Report]

Overall comments:

Thank you for addressing all comments. One more step is needed to accept this manuscript for publication from my end. Hope the authors will address the following suggestions to improve the quality of this manuscript:

For 125-131, could you please check the grammar and sentence structure? If needed, make it several sentences instead of one. For example, “who were responsible for administering the classroom-based survey,” and so on- here, “who” represents two authors, or five members are unclear. In addition, “ensuring confidentiality” will be correct instead of “ensure confidentiality”.

For 2.2 Measures:

1. You have cited that PHQ-9, GAD-7 were used for Bangladeshi population. Could you please include additional information; for example, did the researchers use the Bangla version or not, or how did they administer these scales?

2. In 177, need to mention anxiety related to GAD.

Discussion:

In 301-304, it will be a good start, if you provide references for these factors. For instance, please support with evidence that other researchers have found these factors as factors associated with suicidal ideation.

In 346-348, hope you will find reference(s) for this claim (In rural areas, …..).

In 357, please restructure the sentence because “during” does not mean anything here.

In 376, mental health issue is an umbrella term that encompasses psychological problems (e.g., depression, anxiety) and suicidal behavior. Hence, mental health issues and suicidal ideation are already associated. I got your point, but you need to rephrase this statement for clarification.

In 408-410, Need to rewrite the statement because of its misleading meaning. How a dead person can accidentally be excluded from the study? Hope you have understood my point.

Thank you.

---

## [Reviewer Report]

Thank you for the opportunity to review this revised manuscript! The topic remains highly compelling, as it tackles an underexplored issue in countries like Bangladesh. I appreciate the revisions made thus far and the authors' receptiveness to feedback. However, I do think a few additional points and clarifications are needed to further refine the paper.

1. Grammar and Style

As previously mentioned, the frequent use of “Besides” as a transition word detracts from the readability of the article. While the authors indicated this issue had been resolved, the word still appears throughout the manuscript (e.g., lines 81, 99, 318, etc). Please revisit these instances and replace “Besides” with alternatives such as “Additionally,” “Furthermore,” or “In addition.”

2. Abstract

Include a summary of the factors associated with suicidal planning and attempts, as these analyses were conducted. If no associations were found, explicitly state this for a more comprehensive abstract (however, I see that there were instances of significance).

3. Citations for Literature Gaps

Lines 90–102 reference gaps in the literature but lack citations to substantiate the claims. Please provide specific citations for the referenced studies to clarify which articles inform these statements.

4. Mental Health and Sleep Measures

In lines 161–183, where mental health and sleep measures are discussed, only one item is provided as an example. Could you list all assessed items and their corresponding response options? Alternatively, consider including this information in a supplemental table to enhance transparency.

5. Sentence Rewording

In lines 204–206, the sentence, “The chi-square and Fisher exact test were carried out to determine the association between the characteristics of the participants and suicidal ideation on the basis of lifetime and past year” is a little awkward and could be rephrased for clarity. Suggested revision:

"Chi-square and Fisher’s exact tests were conducted to examine associations between participant characteristics and lifetime or past-year suicidal ideation."

6. Supplementary Table vs. Main Text

In line 210, you mention that additional analyses for suicide planning and attempts are in the supplementary file. Since these findings are equally important, I recommend including them in the main manuscript. Consider either:

Creating a single, larger table to present findings for suicidal ideation, planning, and attempts.

Alternatively, use two tables: one for lifetime findings and another for past-year findings, if this organization is more practical.

7. Handling Missing Data

In line 211, you state that “Any cases with incomplete data on the outcome variable were excluded from the analysis. Missing data in the logistic regression was handled using listwise deletion.” However, the percentage of missing data and the variables affected are unclear. Please clarify by adding a sentence such as:

“Around X% to X% of data was missing, primarily in variables such as [list variables]. This missing data was handled using listwise deletion....” or clarify this in a supplemental file as you deem appropriate.

8. Odds Ratio Interpretation (Lines 283)

Please revise the interpretation of the odds ratio in line 283 as follows:

“Rural adolescents had 25 times the odds of past-year suicidal ideation compared to urban adolescents.”

This phrasing is more accurate as the odds were approximately 25 times, not “over 25 times.” Also, there is an extra “s” at the end of “factor” at the beginning of the line, which should be singular.

9. Discussion Focus

The discussion predominantly focuses on suicidal ideation while giving less attention to planning and attempts. Since all three behaviors are examined in the paper, I recommend expanding the discussion to include a balanced focus on ideation, planning, and attempts.

10. General Feedback

Otherwise, excellent work! The manuscript tackles an important and underexplored topic, and I commend the efforts made so far.

---

## [Editor Report]

Dear authors, The reviewers agree that your work is good and the manuscript should be published, with some minor corrections, which we ask you to review and make.

Reviewer 1:

For 125-131, could you please check the grammar and sentence structure? If needed, make it several sentences instead of one. For example, “who were responsible for administering the classroom-based survey,” and so on- here, “who” represents two authors, or five members are unclear. In addition, “ensuring confidentiality” will be correct instead of “ensure confidentiality”.

For 2.2 Measures:

1. You have cited that PHQ-9, GAD-7 were used for Bangladeshi population. Could you please include additional information; for example, did the researchers use the Bangla version or not, or how did they administer these scales?

2. In 177, need to mention anxiety related to GAD.

Discussion:

In 301-304, it will be a good start, if you provide references for these factors. For instance, please support with evidence that other researchers have found these factors as factors associated with suicidal ideation.

In 346-348, hope you will find reference(s) for this claim (In rural areas, …..).

In 357, please restructure the sentence because “during” does not mean anything here.

In 376, mental health issue is an umbrella term that encompasses psychological problems (e.g., depression, anxiety) and suicidal behavior. Hence, mental health issues and suicidal ideation are already associated. I got your point, but you need to rephrase this statement for clarification.

In 408-410, Need to rewrite the statement because of its misleading meaning. How a dead person can accidentally be excluded from the study? Hope you have understood my point.

Reviewer 2:

1. Grammar and Style

As previously mentioned, the frequent use of “Besides” as a transition word detracts from the readability of the article. While the authors indicated this issue had been resolved, the word still appears throughout the manuscript (e.g., lines 81, 99, 318, etc). Please revisit these instances and replace “Besides” with alternatives such as “Additionally,” “Furthermore,” or “In addition.”

2. Abstract

Include a summary of the factors associated with suicidal planning and attempts, as these analyses were conducted. If no associations were found, explicitly state this for a more comprehensive abstract (however, I see that there were instances of significance).

3. Citations for Literature Gaps

Lines 90–102 reference gaps in the literature but lack citations to substantiate the claims. Please provide specific citations for the referenced studies to clarify which articles inform these statements.

4. Mental Health and Sleep Measures

In lines 161–183, where mental health and sleep measures are discussed, only one item is provided as an example. Could you list all assessed items and their corresponding response options? Alternatively, consider including this information in a supplemental table to enhance transparency.

5. Sentence Rewording

In lines 204–206, the sentence, “The chi-square and Fisher exact test were carried out to determine the association between the characteristics of the participants and suicidal ideation on the basis of lifetime and past year” is a little awkward and could be rephrased for clarity. Suggested revision:

"Chi-square and Fisher’s exact tests were conducted to examine associations between participant characteristics and lifetime or past-year suicidal ideation."

6. Supplementary Table vs. Main Text

In line 210, you mention that additional analyses for suicide planning and attempts are in the supplementary file. Since these findings are equally important, I recommend including them in the main manuscript. Consider either:

Creating a single, larger table to present findings for suicidal ideation, planning, and attempts.

Alternatively, use two tables: one for lifetime findings and another for past-year findings, if this organization is more practical.

7. Handling Missing Data

In line 211, you state that “Any cases with incomplete data on the outcome variable were excluded from the analysis. Missing data in the logistic regression was handled using listwise deletion.” However, the percentage of missing data and the variables affected are unclear. Please clarify by adding a sentence such as:

“Around X% to X% of data was missing, primarily in variables such as [list variables]. This missing data was handled using listwise deletion....” or clarify this in a supplemental file as you deem appropriate.

8. Odds Ratio Interpretation (Lines 283)

Please revise the interpretation of the odds ratio in line 283 as follows:

“Rural adolescents had 25 times the odds of past-year suicidal ideation compared to urban adolescents.”

This phrasing is more accurate as the odds were approximately 25 times, not “over 25 times.” Also, there is an extra “s” at the end of “factor” at the beginning of the line, which should be singular.

9. Discussion Focus

The discussion predominantly focuses on suicidal ideation while giving less attention to planning and attempts. Since all three behaviors are examined in the paper, I recommend expanding the discussion to include a balanced focus on ideation, planning, and attempts.

10. General Feedback

Otherwise, excellent work! The manuscript tackles an important and underexplored topic, and I commend the efforts made so far.

---

## [Editor Report]

After two rounds of revisions, the manuscript is quite good and represents a significant contribution to the knowledge of a problem that is very little studied in LMICs.